# Using Machine Learning for Analysis of Wideband Acoustic Immittance and Assessment of Middle Ear Function in Infants

**DOI:** 10.3390/audiolres15020035

**Published:** 2025-03-31

**Authors:** Shan Peng, Yukun Zhao, Xinyi Yao, Huilin Yin, Bei Ma, Ke Liu, Gang Li, Yang Cao

**Affiliations:** 1Department of Audiology and Speech Language Pathology, Department of Otorhinolaryngology-Head & Neck Surgery, West China Hospital of Sichuan University, Chengdu 610041, China; 2Key Laboratory of Bio-Resource and Eco-Environment of Ministry of Education, College of Life Sciences, Sichuan University, Wangjiang Road 29, Chengdu 610065, China; 3Health Examination Center, Sichuan Electric Power Hospital, Chengdu 610011, China

**Keywords:** machine learning, wideband acoustic immittance, middle ear function, hearing of infants

## Abstract

Objectives: Evaluating middle ear function is essential for interpreting screening results and prioritizing diagnostic referrals for infants with hearing impairments. Wideband Acoustic Immittance (WAI) technology offers a comprehensive approach by utilizing sound stimuli across various frequencies, providing a deeper understanding of ear physiology. However, current clinical practices often restrict WAI data analysis to peak information at specific frequencies, limiting its comprehensiveness. Design: In this study, we developed five machine learning models—feedforward neural network, convolutional neural network, kernel density estimation, random forest, and support vector machine—to extract features from wideband acoustic immittance data collected from newborns aged 2–6 months. These models were trained to predict and assess the normalcy of middle ear function in the samples. Results: The integrated machine learning models achieved an average accuracy exceeding 90% in the test set, with various classification performance metrics (accuracy, precision, recall, F1 score, MCC) surpassing 0.8. Furthermore, we developed a program based on ML models with an interactive GUI interface. The software is available for free download. Conclusions: This study showcases the capability to automatically diagnose middle ear function in infants based on WAI data. While not intended for diagnosing specific pathologies, the approach provides valuable insights to guide follow-up testing and clinical decision-making, supporting the early identification and management of auditory conditions in newborns.

## 1. Introduction

Newborn hearing screening plays a pivotal role in detecting and addressing hearing issues in infants at an early stage [1]. Evaluating middle ear function is a crucial aspect of this screening process, as it influences the choice of intervention strategies for children with hearing impairments [2]. Furthermore, middle ear disease, such as otitis media, stands as one of the most prevalent childhood illnesses globally, affecting millions of infants annually [3]. While manageable in many cases, untreated instances can lead to severe complications like hearing loss, delayed speech and language development, and even cognitive impairment [4]. Hence, early assessment of middle ear status is imperative to mitigate the impact of hearing issues on infant health and development.

Conventional diagnostic techniques for assessing middle ear status, like pneumatic otoscopy and tympanometry, come with limitations in accuracy and reliability, especially when dealing with infants and young children [5]. The single-frequency tympanogram tests typically used in clinical settings, such as 226 Hz and 1000 Hz tympanometry, fall short in fully capturing the middle ear function status in children [6]. Wideband acoustic impedance (WAI) measurement, offering a non-invasive and objective approach, has emerged as a promising tool for assessing middle ear status [6,7]. WAI provides comprehensive insights into the biomechanical properties of the middle ear, encompassing compliance, stiffness, and damping characteristics. This enables more precise diagnosis and monitoring of middle ear diseases [8]. However, despite its potential, the clinical analysis of WAI data remains insufficiently comprehensive, often focusing only on peak information from specific frequencies, which restricts its diagnostic utility.

Recent advancements in machine learning (ML) techniques have revolutionized medical diagnostics by enhancing the accuracy and efficiency of disease prediction models [9]. By leveraging complex algorithms and large datasets, ML algorithms can uncover patterns and relationships within WAI data that may not be readily discernible through traditional analysis methods [10]. Furthermore, ML models can adapt and improve their predictive capabilities over time, making them valuable assets in the clinical assessment of middle ear health in infants. Additionally, ML models can adapt and improve their predictive capabilities over time, enhancing clinical assessment of middle ear health in infants. Although research has applied ML to diagnose middle ear infections, it has primarily focused on age groups older than 1 year [11,12]. Assessments of middle ear function in younger infants under one year of age have not yielded conclusive results. Furthermore, there remains a lack of comprehensive and automated approaches to enhance diagnostic accuracy and improve clinical efficiency in this critical age group.

This study aims to address these gaps by evaluating the effectiveness of ML in analyzing WAI data to predict middle ear function in infants. We conduct a thorough evaluation of various ML algorithms to assess their accuracy in identifying middle ear function. Additionally, we develop an interactive GUI program for processing WAI data of infants. The primary goal of this work is to support the interpretation of hearing screening results and guide follow-up testing, thereby contributing to the early identification and management of auditory conditions in pediatric populations.

## 2. Methods

### 2.1. Participants

A total of 468 ears of infants who visited the Hearing Center of West China Hospital, Sichuan University from January 2021 to June 2023 were included in this study for the collection of Wideband Acoustic Immittance (WAI) data (Table 1). Prior to WAI evaluation, all infants underwent an otoscopic examination to exclude those with conditions that could interfere with the measurements. Exclusion criteria included perforation of the tympanic membrane, otitis media, obstruction of the external auditory canal, or any other condition that constituted an impediment to WAI testing. Infants with normal middle ear function met the following criteria: passing the Distortion Product Otoacoustic Emission (DPOAE) or Transient Evoked Otoacoustic Emission (TEOAE) screening test, and exhibiting a single positively peaked tympanogram at 1000 Hz tympanometry. In contrast, infants with abnormal middle ear function were identified by the following criteria: abnormalities in the DPOAE or TEOAE screening test, and the absence of a peak in the 1000 Hz tympanogram [13]. Based on these criteria, 357 ears were identified as having normal middle ear function, while the remaining 111 ears were categorized as exhibiting abnormal middle ear function (Table 1).

This study was approved by the Ethics Committee on Biomedical Research, West China Hospital of Sichuan University (Approval No. 2022 (145); approval date: 29 January 2022). Parental consent was obtained before the test.

### 2.2. WAI Data Processing

Each sample was tested using the Titan hearing test platform (Interacoustics, Denmark) with the default test parameters of the device, using a broadband click as the probe tone. Each file contains crucial information as follows:

FREQ: Consists of 107 stimulus frequency points ranging from 226 to 8000 Hz, consistent across all samples.

PRESSURE: Encompasses measured sound pressure values from −400 to +200 daPa. This varies among samples, with most having around 50 data points and a few having very few data points.

ABSORBANCE: Holds the absorbance corresponding to each frequency and sound pressure, represented as a two-dimensional matrix. The rows signify the sound pressures, while the columns represent the frequencies.

TYMP_Y_226 Hz and TYMP_Y_1000 Hz: Contain the ear sample’s admittance values at different sound pressure values for 226 Hz and 1000 Hz, respectively, measured in mmho.

Additionally, the original data include basic sample information and machine operation details. We developed a program to extract frequency, pressure, and absorbance data for each sample. Due to variations in the number and step sizes of sound pressure values per sample (ranging from 40 to 60), the inconsistent feature count (absorbance matrix) impedes subsequent machine learning processes. Hence, standardizing the WAI data is necessary.

We employed a range of −350 to +150 daPa for sound pressure, excluding 8 samples with values falling outside this range. For pressure and corresponding absorbance at each frequency, we utilized piecewise cubic Hermite interpolation (PCHIP) with a uniform step size of 10 daPa. During this process, we identified and excluded 58 samples with duplicate sound pressure values causing errors. Post-interpolation, each sample comprises 51 sound pressure values, resulting in 5457 features (107 × 51) per sample. A heatmap comparison of absorbance pre- and post-interpolation is provided. After final data filtering, there are 94 samples with middle ear abnormalities and 308 samples with normal middle ears. Python (version 3.10.11), with emphasis on the NumPy (version 1.25.0) and pandas (version 1.5.3) scientific computing libraries, was employed for this procedure. Absorbance data for each sample were extracted in Dataframe format and stored in a CSV file.

### 2.3. Predictive Model Construction

Predictive models are increasingly vital across numerous medical disciplines for purposes such as diagnostics and prognostics [14]. They are developed by learning “experience”, which refers to data collected from real patient cases to promote the analysis of various kinds of complex data [15,16,17]. In this work, we employed the popular machine learning models, including feedforward neural network, convolutional neural network, kernel density estimation, random forest, and support vector machine, to predict and assess the normalcy of middle ear function.

#### 2.3.1. Neural Network Classification Model

The neural network model is composed of numerous neurons, each functioning as a mathematical unit to process input data. Organized in a hierarchical structure, the model comprises input, hidden, and output layers. The input layer receives raw data, the hidden layers identify patterns and features by performing calculations, and the output layer produces the final result, such as a classification or prediction.

For this study, we used a straightforward four-layer feedforward neural network (FNN) designed specifically for the relatively simple dataset we analyzed. This model takes 5457 features extracted from the WAI data as input and classifies each sample into one of two categories: either the presence of middle ear pathology (indicated by ‘1’) or normal middle ear function (indicated by ‘0’). The structure of the feedforward neural network is illustrated in Figure 1.

During training, the model learns by minimizing errors using a method called binary cross-entropy (BCE) loss. To guide this learning process, we set the initial learning rate at 0.001 and used the Adam optimizer, which is a commonly used method for adjusting the model’s learning steps. Training was conducted in small groups of 32 samples (batch size) over 100 complete passes through the dataset (epochs). The model was built and trained using the PyTorch (version 2.0.1) library.

The feedforward neural network (FNN) classifier starts with an input layer that takes 5457 features from the data. It then passes the information through three layers of interconnected nodes (hidden layers), which contain 1000, 500, and 100 nodes, respectively. These layers use a function called ReLU to activate the nodes and identify patterns in the data. Finally, the output layer, consisting of a single node, uses a sigmoid function to convert the processed data into a probability for classifying each sample as either normal or indicating middle ear pathology.

#### 2.3.2. Convolutional Neural Network Classification Model

Convolutional neural networks (CNNs) are a class of feedforward neural networks characterized by their incorporation of convolutional computations and depth. Comprising one or more convolutional layers followed by fully connected layers (analogous to classic neural networks), CNNs also include associated weights and pooling layers. This architecture enables CNNs to effectively leverage the two-dimensional structure of input data. Compared to other deep learning architectures, CNNs have demonstrated superior performance in tasks such as image and speech recognition. The trained CNNs were able to classify paired auditory brainstem response (ABR) waveforms with ideal accuracy, sensitivity, and specificity [18].

In this study, the WAI data consist of absorbance matrices at different sound pressures and frequencies, inherently two-dimensional in nature. Each sample’s WAI data are treated as a grayscale image with a single channel, with a resolution of 51 × 107 (number of sound pressures × number of frequencies). Absorbance values are converted to grayscale intensities. The task of the CNN model is to classify each sample’s WAI image as either normal or abnormal. The framework of the CNN model is illustrated (Figure 2).

During input preprocessing, to facilitate convolutional operations, a row and column of zeros are added to the absorbance matrix, increasing its resolution to 52 × 108. After two rounds of convolution (Conv) and pooling (Pool), the size of the feature map becomes 16 × 13 × 27. Following this, the feature map is flattened and mapped to a two-dimensional output vector through fully connected layers (FC), and finally, a softmax layer outputs the probability of each type of classification.

During training, the model learns by minimizing errors using binary cross-entropy (BCE) loss. We set the learning speed (learning rate) at 0.001 and used the Adam optimizer to adjust the learning process. Training was conducted in small groups of 32 samples (batch size) over 100 complete cycles through the data (epochs).

The CNN comprises two convolutional layers followed by max-pooling operations and ReLU activation functions. The first convolutional layer applies 8 filters of size 3 × 3, producing feature maps of dimensions B × 8 × 52 × 108, which are then downsampled via max-pooling to B × 8 × 26 × 54. The second convolutional layer employs 16 filters of size 3 × 3 on the pooled feature maps, yielding feature maps of dimensions B × 16 × 26 × 54, which are again downsampled to B × 16 × 13 × 27. These feature maps are flattened and fed into a fully connected layer, followed by softmax activation to output class probabilities.

#### 2.3.3. Kernel Density Estimation Classification Model

Kernel density estimation (KDE) is a non-parametric method for estimating the probability density function behind the data. It has been applied in analyzing complex physiological data, such as miRNA markers of forensic body fluid, direct electrical stimulation (DES) data in neurosurgical patients, and surface electromyographic (sEMG) signals related to myoelectric control [16,19,20]. It estimates the probability density function by placing kernel functions around each data point and then smoothing and weighting these kernel functions to obtain the final estimate of the data’s probability density [21]. For a dataset containing *n* samples, the probability density of a sample *x* (*ρ*) is calculated using the following formula:$$\rho \left(x\right)=\frac{1}{nh}\sum_{i=1}^{n}K\left(\frac{x-x_{i}}{h}\right),i\in \left[1\;,\;n\right]$$

In this task, we employ the radial basis function (RBF) as the kernel function K, defined by the following formula:Ky=12πⅇ−y22, y=x−xih

For each input sample, we compute its probability density under each class, thereby predicting whether it belongs to the category of diseased or normal middle ears. In density estimation, the bandwidth parameter h controls the width of the kernel function, which significantly affects the results of kernel density estimation. A smaller bandwidth may result in overly detailed estimates, potentially leading to overfitting, while a larger bandwidth may result in overly smooth estimates, potentially leading to underfitting. To select an appropriate bandwidth, we tested a range of bandwidth values from 0.1 to 8.0 for their effect on predicting WAI data classification using KDE. We found that a bandwidth value of 4.0 yielded the best performance across all metrics. Therefore, we chose 4.0 as the model’s bandwidth. The KDE model algorithm is implemented by using Python, and the trained KDE model is saved and named according to the pickle library.

#### 2.3.4. Random Forest Classification Model

Random forest (RF) is an ensemble learning method commonly used for classification and regression problems. It consists of multiple decision trees, where each decision tree is a fundamental model composed of nodes and edges. Starting from the root node, the tree recursively splits based on features, eventually reaching leaf nodes and providing corresponding predictions. Classification predictions are made by aggregating the results of these decision trees through voting or averaging. Random forests offer many advantages, such as handling high-dimensional data, large training datasets, and mitigating overfitting. Additionally, they provide insights into feature importance, aiding in feature selection and model interpretation. RF has been widely used in audiology, including analyzing the relationship between age-related hearing loss and speech recognition decline, predicting individualized hearing aid benefits, and assessing ototoxicity in patients undergoing radiation therapy [22,23,24].

In this study, we utilize the RandomForestClassifier class from the sklearn.ensemble library to construct the random forest classifier object with the following parameter settings: *n*_estimators = 100, specifying the number of decision trees as 100; criterion = ‘gini’, measuring node purity using the Gini impurity criterion; min_samples_split = 2, setting the minimum number of samples required to split an internal node to 2; and min_samples_leaf = 1, setting the minimum number of samples required to be at a leaf node to 1. The trained RF model is saved and called using the pickle library.

#### 2.3.5. Support Vector Machine Classification Model

Support vector machine (SVM) is a widely used supervised learning algorithm primarily used for classification and regression problems. In classification tasks, SVM identifies an optimal hyperplane to separate samples from different classes. Based on this hyperplane, SVM determines a decision boundary to classify samples into different categories. For new unlabeled samples, SVM predicts their class by computing their projection onto the optimal hyperplane. The sample’s projection can be used to determine its class membership.

In this study, we employ the SVC class from the sklearn.svm library to construct the support vector machine classifier object with the following parameter settings: C = 1.0, the regularization parameter controlling the balance between classification errors and decision boundaries; set 1.0. kernel = ‘rbf’, kernel function type set to radial basis function; gamma = ‘scale’, kernel parameter for radial basis function automatically calculated based on the number of features; probability = True, enabling probability estimation to obtain the probability of samples belonging to each class; and class_weight = none, preventing the model from automatically adjusting weights and used for handling imbalanced datasets. The trained SVM model is saved and called using the pickle library.

### 2.4. Training and Testing

Due to the limited sample size (just over 300), ten-fold cross-validation is employed to evaluate the performance and generalization ability of the models. The original dataset is divided into 10 equal subsets. Nine of these subsets are used for training the models, while the remaining subset is used for testing each classification model. This process yields evaluation metrics such as accuracy, precision, recall, F1 score, and MCC (Matthews Correlation Coefficient) for each model. This process is repeated 10 times, with a different subset chosen as the test set each time. Each type of model ultimately obtains 10 independent model evaluation results.

For each model type, the model with the best performance on a selective fold is integrated into the final prediction program. The sklearn.model_selection library is utilized for fold separation, the sklearn.metrics library for calculating evaluation metrics, and the matplotlib library for line plot generation.

### 2.5. Control Sample Searching and Plotting

To assist in judgment, a control sample with the opposite predicted label is sought from the training samples. This control sample is most similar to the input sample. Both the input sample and the control sample are then plotted as absorbance heatmaps displayed below the output box. The cosine similarity is used to judge the similarity between the feature vectors of the two samples. The matplotlib library is used for plotting absorbance heatmaps.

### 2.6. Tympanogram Admittance Curve Plotting

For further assistance in judgment, the numerical values of TYMP_Y_226 Hz and TYMP_Y_1000 Hz are combined with sound pressure values. Using the matplotlib library, tympanogram admittance curves are plotted, and the scipy.signal library’s find_peaks function is utilized for peak identification in the curves. The prominence threshold for peak identification is set to 0.15.

### 2.7. Program Integration and GUI Development

The pyinstaller library is employed to package all dependencies and interpreters into standalone executable files for use on different devices. To reduce program size, use PyTorch libraries. Moreover, pth model files are converted to ONNX format model files using the ONNX library. After packaging, the program, along with the model, occupies a space of no more than 150 MB. The PySimpleGUI library is used to create an interactive GUI interface.

## 3. Results

### 3.1. Performances of Predictive Models

For predictive models, the following five metrics are used to evaluate their performance, where TP stands for true positive, TN stands for true negative, FP stands for false positive, and FN stands for false negative:Accuracy: Accuracy = (TP + TN)/(TP + TN + FP + FN).Precision: Measures the proportion of true positive samples among those predicted as positive by the model. Precision = TP/(TP + FP).Recall: Measures the proportion of true positive samples successfully identified by the model among all positive class samples. Recall = TP/(TP + FN).F1 Score: Provides a composite measure of model performance considering both precision and recall. F1 Score = 2 × Precision × Recall/(Precision + Recall).Matthews Correlation Coefficient (MCC): Used to measure the performance of binary classification models, especially suitable for imbalanced data. MCC = (TP × TN − FP × FN)/sqrt((TP + FP) × (TP + FN) × (TN + FP) × (TN + FN)).

We performed ten-fold cross-validation, using 362 samples for training and 40 for testing, to assess the performance of five predictive models as shown below.

#### 3.1.1. Feedforward Neural Network Model (FNN)

The FNN model exhibited overall stability during cross-validation, with only occasional instances of instability observed across multiple cross-validation runs (occurring once in ten-fold cross-validation) (Figure 3). Its predictive performance was satisfactory, with all the metrics maintaining an average value above 0.8. The model trained in the first fold demonstrated the best overall performance, leading to its selection for integration into the program.

#### 3.1.2. Convolutional Neural Network Model (CNN)

The CNN model demonstrated stable performance across all folds of cross-validation. Its predictive efficacy slightly exceeded that of the simple feedforward neural network, representing the highest average metrics among all models (Figure 4). The model trained on the seventh fold exhibited the best overall performance and was chosen for integration into the program.

#### 3.1.3. Kernel Density Estimation Model (KDE)

The KDE classification model showed stable performance throughout cross-validation, without any instances of extremely low metrics (Figure 5). Its predictive efficacy was comparable to that of neural network models. The model trained on the first fold exhibited the best overall performance and was integrated into the program.

#### 3.1.4. Random Forest Model (RF)

The RF model’s performance closely approximated that of neural network and kernel density estimation models (Figure 6). The model trained on the first fold displayed the best overall performance and was integrated into the program.

#### 3.1.5. Support Vector Machine Model (SVM)

The SVM model demonstrated excellent predictive performance, exhibiting the highest precision among all models (Figure 7). Its average metrics were only slightly inferior to those of the convolutional neural network model. The model trained on the first fold displayed the best overall performance and was integrated into the program.

Overall, the performance of all five models in the test set was satisfactory, with average values of all metrics exceeding 0.8. The difference in performance between the best-performing model and other models was less than 5%.

### 3.2. Usage of the Integrated Program

As shown in Figure 8, click on the button to the right of the input box to select the original WAI file for middle ear function prediction (sample_name.m). You can also choose the output path for the result file.

Once the sample selection is completed, click on the predict button. After the program finishes running, as shown in the figure below, sample information, the judgment results of each model, absorbance heatmaps of the input sample and the contrast sample, tympanogram admittance curves at 226 Hz and 1000 Hz, and peak identification will be displayed as below. Additionally, a text version of the result file (output.txt) will be outputted to the user-specified path.

For the input sample, the leftmost column (METHOD) displays the prediction model used. Each model will provide a judgment (PREDICTION column: ‘1’ indicating diseased, ‘0’ indicating normal), along with the probability value of the judgment (PROBABILITY column). A probability value closer to 1 indicates higher confidence, while a value close to 0.5 suggests the prediction is close to random. The program calculates the classification outputs of each model, and if more than half of the models (=3 models) provide a normal classification, the input sample is classified as normal; otherwise, it is classified as abnormal. If two or more models provide probability values ≤ 0.8, the program will prompt the user that the prediction for the sample may not be accurate and further examination is needed to determine if it is abnormal.

Below the prediction results are the absorbance heatmaps of the input sample and the contrast sample, aiding in identifying abnormalities by visualizing the similarities. Below the heatmaps are the tympanogram admittance curves at 226 Hz and 1000 Hz. The program automatically performs peak identification (marked with a red “x”) and classifies the tympanogram at 226 Hz as type A (peaked), B (flat), or C (negative pressure). The peak detection results are also displayed below the 1000 Hz graph.

In clinical practice, the inclusion criteria for middle ear abnormalities are (1) 226 Hz tympanogram: type B or C; (2) 1000 Hz tympanogram: no peak; (3) OAE not passed. In this program, if there is a contradiction between the tympanogram curves and the integrated model judgment results, namely:(1)The model judges the sample as abnormal, but the 226 Hz tympanogram is type A or there is a peak in the 1000 Hz tympanogram.(2)The model judges the sample as normal, but the 226 Hz tympanogram is not type A, and there is no peak in the 1000 Hz tympanogram.

In these two cases, the program will prompt the user that the prediction for the sample may not be accurate, and further examination is needed to determine if it is abnormal, as shown in Figure 9.

After selecting a file each time, the program will automatically check the input file. If the input file is not WAI data or if there are issues with the WAI data, the user will be prompted to make modifications, as shown in Figure 10.

## 4. Discussion

The utilization of machine learning (ML) techniques in the analysis of wideband acoustic immittance (WAI) data presents a promising avenue for the assessment of middle ear function in infants. Our study demonstrates the feasibility and effectiveness of employing various ML models for automatic diagnosis, offering substantial improvements over conventional methods.

The developed ML models demonstrated robust performance, with accuracy exceeding 90% and other metrics consistently above 0.8, highlighting their clinical potential. The diversity of models employed, including feedforward neural networks, convolutional neural networks, kernel density estimation, random forest, and support vector machines, provides flexibility and adaptability to different datasets and scenarios. Notably, the convolutional neural network model exhibited the highest average metrics, suggesting its efficacy in capturing complex patterns within WAI data.

One of the key advantages of our approach is the comprehensive analysis of WAI data across multiple frequencies, enabling a deeper understanding of middle ear physiology. Traditional clinical practices often focus on peak information at specific frequencies, which may overlook subtle abnormalities or variations in ear condition [25]. By leveraging ML algorithms to extract features from wideband data, our models can discern nuanced patterns indicative of middle ear pathology, thereby enhancing diagnostic accuracy and clinical decision-making after newborn hearing screening. These findings are consistent with previous research, which has highlighted the potential of ML in WAI interpretation [11,12,26].

The development of an integrated program with a user-friendly graphical interface facilitates seamless application of ML models in clinical settings. The program enables clinicians to input WAI data and obtain automated predictions of middle ear status, accompanied by visualizations of tympanogram admittance curves and absorbance heatmaps for enhanced interpretation. Furthermore, the program incorporates validation checks to alert users of potential discrepancies between model predictions and tympanogram findings, promoting cautious interpretation and follow-up assessment as necessary.

Our study highlights the feasibility and effectiveness of using machine learning models for the analysis of WAI data in infants. Unlike conventional methods that rely on specific frequencies such as 226 Hz tympanometry, the ML models can leverage the depth of information available from WAI measurements across a wide frequency range. This comprehensive approach may provide clinicians with a more nuanced understanding of middle ear function and help differentiate between various conductive pathologies in cases where conventional methods might be less reliable.

While our study demonstrates promising results, several limitations warrant consideration. First, the interpretation of our results requires awareness that the classification of middle ear function in this study was based solely on acoustic impedance measurements, such as tympanometry and otoacoustic emissions. These non-invasive methods are commonly used in clinical practice to evaluate middle ear function but do not involve direct evaluation or confirmation of middle ear pathology through methods like otomicroscopy or surgery [27,28]. Although we used widely accepted non-invasive methods like 226 Hz and 1000 Hz tympanometry as reference standards, these methods have reliability limitations. Future studies should incorporate clinical assessments such as otomicroscopy or surgical findings, where possible, to provide more robust validation of the models’ predictive accuracy in diagnosing middle ear pathologies. Additionally, although the sample size used in this study is reasonable given the challenges of obtaining high-quality WAI data from infants, the dataset utilized for model training and evaluation primarily comprises infants aged 2–6 months, limiting generalizability to other age groups or populations. Future research should explore the applicability of ML-based WAI analysis across a broader spectrum of ages and clinical contexts with larger sample sizes, including children and adults with diverse hearing profiles and pathologies. Furthermore, ongoing refinement and optimization of ML algorithms are essential to enhance model performance and address challenges such as data variability and class imbalance. Future strategies could include data augmentation techniques to increase sample diversity and sophisticated resampling methods to mitigate class imbalance and improve model robustness.

While this study primarily aimed to detect general middle ear function abnormalities to aid in interpreting newborn hearing screening results, future work could extend these ML models to differentiate specific conductive pathologies. For example, WAI data have been valuable in diagnosing conditions like stapes fixation [29], superior canal dehiscence [30], and partial effusions [31]. By refining the ML algorithms to recognize patterns specific to these conditions, it may be possible to develop models that not only detect middle ear abnormalities but also categorize them into more specific pathologies, offering even greater clinical utility.

In conclusion, the integration of machine learning techniques into the analysis of wideband acoustic immittance data offers a valuable tool for the assessment of middle ear function in infants. Our study underscores the potential of ML models to automate diagnosis, improve accuracy, and facilitate early intervention in middle ear diseases, thereby contributing to enhanced auditory health outcomes for pediatric populations. Further research and clinical validation are warranted to realize the full potential of this approach in routine clinical practice.

## Figures and Tables

**Figure 1 audiolres-15-00035-f001:**
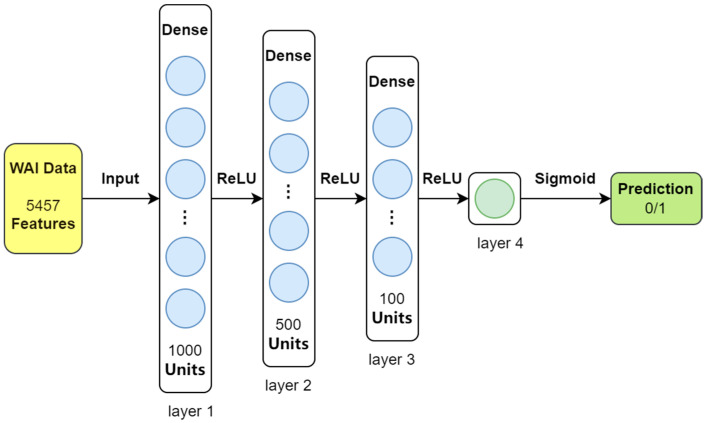
Feedforward neural network classification model.

**Figure 2 audiolres-15-00035-f002:**
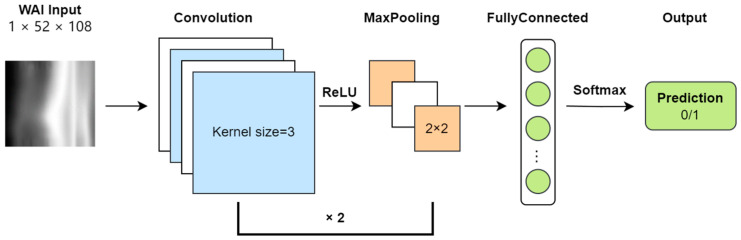
Convolutional neural network classification model.

**Figure 3 audiolres-15-00035-f003:**
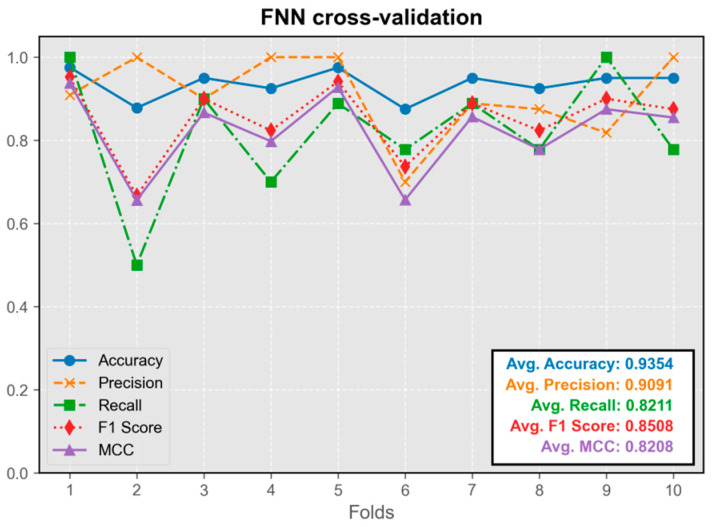
Cross-validation results of the FNN model.

**Figure 4 audiolres-15-00035-f004:**
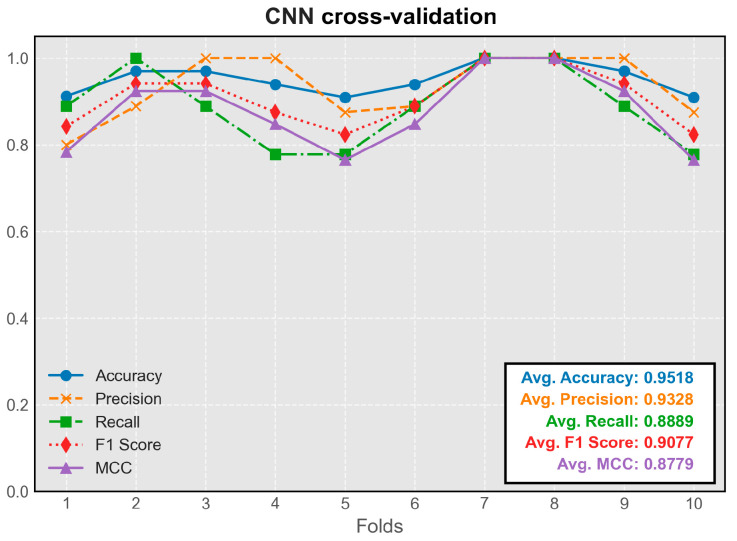
Cross-validation results of the CNN model.

**Figure 5 audiolres-15-00035-f005:**
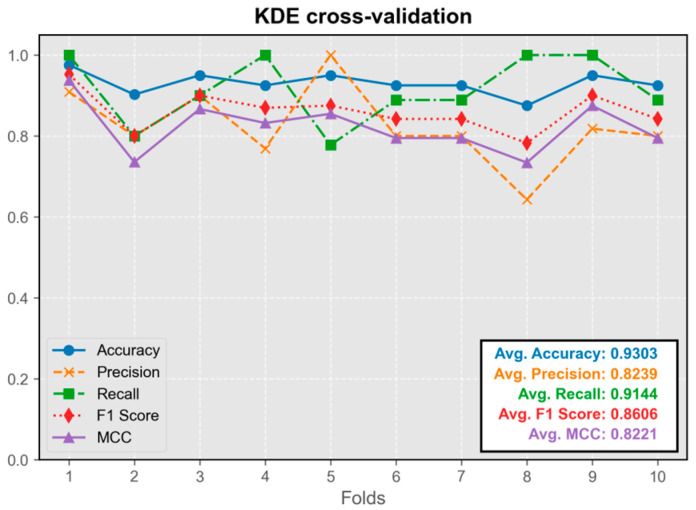
Cross-validation results of the KDE model.

**Figure 6 audiolres-15-00035-f006:**
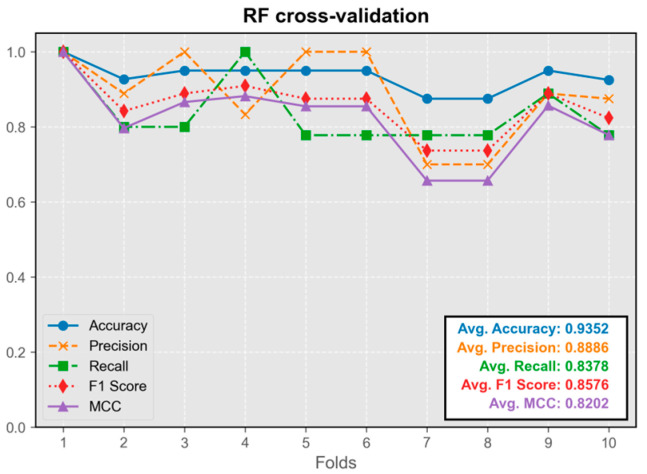
Cross-validation results of the RF model.

**Figure 7 audiolres-15-00035-f007:**
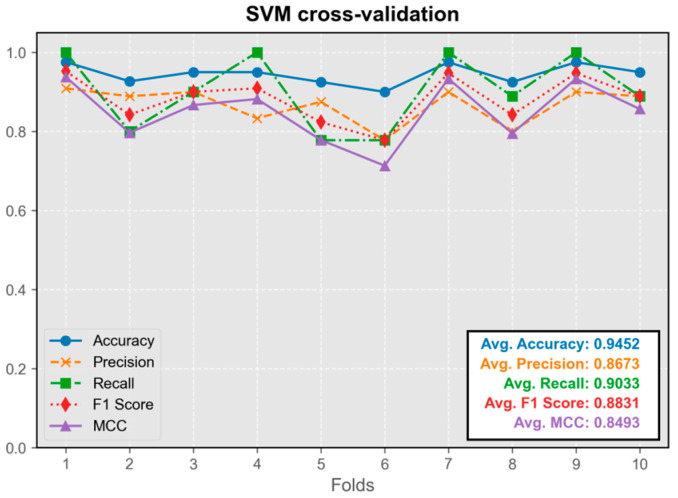
Cross-validation results of SVM model.

**Figure 8 audiolres-15-00035-f008:**
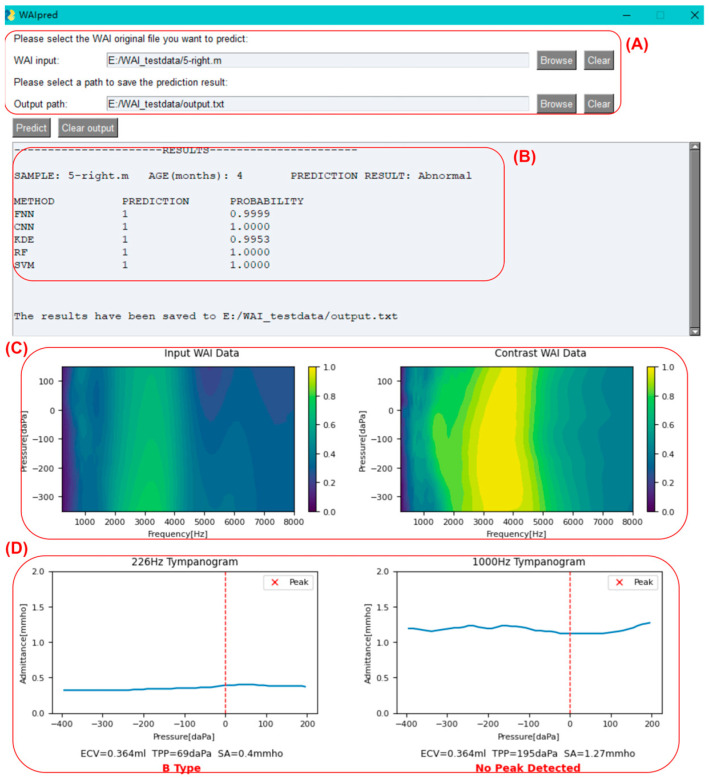
Software interface display of WAIpred. (**A**) WAI file input area; (**B**) text results display area, including basic input information and the prediction results for each model; (**C**) visualization of the input WAI data and contrast WAI data; and (**D**) auxiliary judgment area, including the admittance curve at 226 Hz and 1000 Hz.

**Figure 9 audiolres-15-00035-f009:**
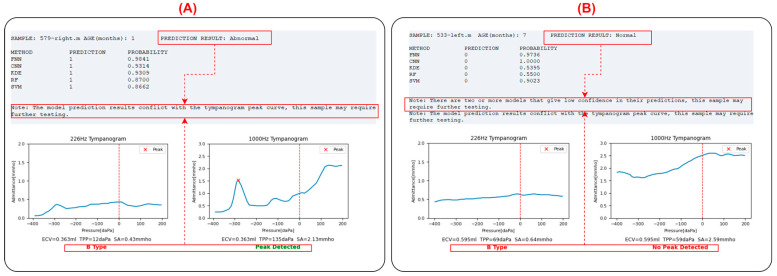
Conflict detection between model judgment results and tympanogram curves. (**A**) The model judges the sample as abnormal, but the 226 Hz tympanogram is type A or there is a peak in the 1000 Hz tympanogram. (**B**) The model judges the sample as normal, but the 226 Hz tympanogram is not type A, and there is no peak in the 1000 Hz tympanogram [13].

**Figure 10 audiolres-15-00035-f010:**
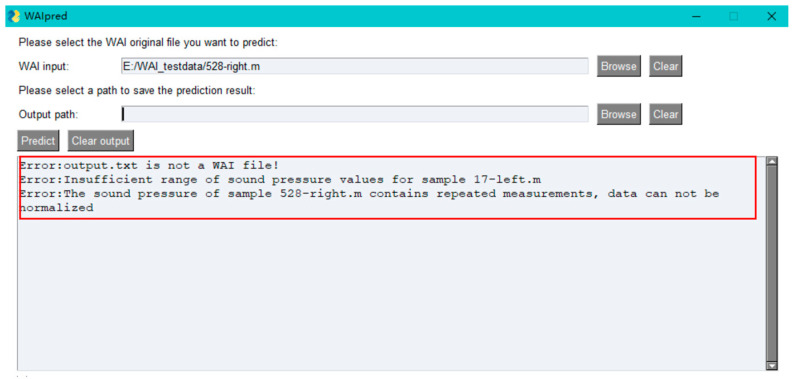
WAI file check display. The red box indicates examples of prompted errors

**Table 1 audiolres-15-00035-t001:** Information on the ears of infants included in this study.

	Male/Female	Total Number of Infants	Age Range (Months, Mean ± SD)	Number of Ears Left/Right	Total Number of Ears
Infants with normal middle ear function	151/89	240	3 ± 2.3	172/185	357
Infants with abnormal middle ear function	39/44	83	5 ± 2.7	57/54	111

## Data Availability

The data presented in this study are available on request from the corresponding author due to privacy concerns.

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
