# Peer review of "Using Machine Learning for Analysis of Wideband Acoustic Immittance and Assessment of Middle Ear Function in Infants"

_audiolres, 2025, doi:10.3390/audiolres15020035_

Round 1
Reviewer 1 Report (New Reviewer)
Comments and Suggestions for Authors
This is a study on diagnostic accuracy of machine learning models based on wideband acoustic impedance in order to assess middle-ear status. The title and abstract are concise and informative. The introduction provides enough background data, discusses study rationale and lists major aims.
The MM section listed inclusion and exclusion criteria and major study outcomes. IRB approval was noted, and informed consent was noted. Drop out ratio was mentioned. The computation methods of ML were well-described, transparent, reproducible and rigorously constructed.
The results were well portryed with tables and figures, and complicated predictive models were simplified enough for the reader to comprehend. The discussion listed major limitations and areas of further improvement, but the results give an interesting view on improving clinical work and offer a decision supporting model that is important to publish and validate further.
Author Response
Thank you very much for taking the time to review this manuscript. We appreciate your valuable comments.
Reviewer 2 Report (New Reviewer)
Comments and Suggestions for Authors
The introduction of the manuscript is commendable and methodical. The methodology you employed is clearly articulated. It would enhance the methodology to support the models in sections 2.3.2 through 2.3.4. This approach will strengthen the documentation and allow you to substantiate the areas included in the protocol. It is crucial to connect the existing knowledge within the model framework to prior studies that utilized a similar methodology. Additionally, your bibliography should be formatted in accordance with the journal's standards, ensuring that the font is appropriately styled and that all missing DOIs are included.
Author Response
We appreciate the reviewer’s thoughtful feedback and constructive suggestions. Below, we address each of the points raised:
Comment 1: Support for Models in Sections 2.3.2 through 2.3.4 and connection to Existing Knowledge
Response 1: We acknowledge the reviewer’s suggestion to enhance the methodology by supporting the models presented in sections 2.3.2 through 2.3.4 and connecting our work to existing knowledge. In response, we have included additional references and explanations connecting these models to prior studies that employed similar methodologies (highlighted in sections 2.3.2 through 2.3.4 of revised manuscript). This modification strengthens the methodological framework by highlighting the models' relevance and application in the context of audiologic data analysis.
Comment 2: Bibliography Formatting
Response 2: We have revised the bibliography to ensure it adheres to the journal’s formatting standards. We have included missing DOIs where applicable. The updated references section now aligns with the journal’s guidelines for submission.
We hope that these revisions address the reviewer’s concerns. Thank you again for your helpful comments.
Reviewer 3 Report (New Reviewer)
Comments and Suggestions for Authors
The manuscript presents a well-structured and methodologically sound study on the application of machine learning for assessing middle ear function in infants using Wideband Acoustic Immittance (WAI) data. The integration of multiple machine learning models enhances the diagnostic accuracy and provides valuable insights for early detection of auditory conditions. The development of a user-friendly software interface further strengthens the clinical applicability of the study. The statistical analysis is appropriate, and the discussion effectively contextualizes the findings while acknowledging the study’s limitations. The writing is clear and professional, with only minor areas that could benefit from slight refinements for clarity and readability.
Overall, this is a well-executed and meaningful contribution to the field of audiology and machine learning.
Congratulations to the authors on their excellent work.
Author Response
Thank you very much for your insightful comments. We have carefully revised the manuscript to improve its clarity and readability as highlighted in the revised manuscript (Line 15-16, Line 65, Line 304-305, Line 404-405). These changes should enhance the flow of the paper. We hope these revisions meet your expectations.
This manuscript is a resubmission of an earlier submission. The following is a list of the peer review reports and author responses from that submission.
Round 1
Reviewer 1 Report
Comments and Suggestions for Authors
Comments on revised version of manuscript - the authors are still comparing one proxy for middle ear function (DPOAE and tympanometry) to another proxy (machine learning based on wideband tympanometry) in infants who never had their middle ear anatomy checked.
As I commented in the previous version this does not yield any new info on how we shall diagnose the middle ear and hearing in small iunfants. The authors need medical data on the eardrum pathologies in these children before any new info can be established.
Reviewer 2 Report
Comments and Suggestions for Authors
Revisions done